# A Modular Metagenomics Pipeline Allowing for the Inclusion of Prior Knowledge Using the Example of Anaerobic Digestion

**DOI:** 10.3390/microorganisms8050669

**Published:** 2020-05-05

**Authors:** Daniela Becker, Denny Popp, Hauke Harms, Florian Centler

**Affiliations:** Department of Environmental Microbiology, Helmholtz Centre for Environmental Research – UFZ, Permoserstraße 15, 04318 Leipzig, Germany; daniela.taraba@ufz.de (D.B.); denny.popp@ufz.de (D.P.); hauke.harms@ufz.de (H.H.)

**Keywords:** metagenomics, microbial communities, compositional analysis, functional profiling, anaerobic digestion

## Abstract

Metagenomics analysis revealing the composition and functional repertoire of complex microbial communities typically relies on large amounts of sequence data. Numerous analysis strategies and computational tools are available for their analysis. Fully integrated automated analysis pipelines such as MG-RAST or MEGAN6 are user-friendly but not designed for integrating specific knowledge on the biological system under study. In order to facilitate the consideration of such knowledge, we introduce a modular, adaptable analysis pipeline combining existing tools. We applied the novel pipeline to simulated mock data sets focusing on anaerobic digestion microbiomes and compare results to those obtained with established automated analysis pipelines. We find that the analysis strategy and choice of tools and parameters have a strong effect on the inferred taxonomic community composition, but not on the inferred functional profile. By including prior knowledge, computational costs can be decreased while improving result accuracy. While automated off-the-shelf analysis pipelines are easy to apply and require no knowledge on the microbial system under study, custom-made pipelines require more preparation time and bioinformatics expertise. This extra effort is minimized by our modular, flexible, custom-made pipeline, which can be adapted to different scenarios and can take available knowledge on the microbial system under study into account.

## 1. Introduction

Modern next-generation sequencing (NGS) methods produce huge amounts of data that need to be analyzed. While the laboratory part of NGS is straightforward thanks to established protocols and instructions provided by the manufacturers, data analysis is an often-underestimated challenge and time intensive. Currently, metagenomics is the state-of-the-art approach to investigate the metabolic potential of microbial communities. The advantage over marker gene-based methods like 16S rRNA gene amplicon sequencing is that, besides the composition, the functional potential of the whole community is unveiled, providing a basis to better understand the dynamics, potential interactions, and ecology of microbial communities [1,2].

Metagenomic short reads can either be directly taxonomically analyzed, by binning or mapping to reference databases [3,4], or subjected to an assembly workflow, which is in the focus of this work. A typical metagenomics assembly workflow consists of several steps. Starting from the demultiplexed reads from which the technical sequences (adapter and barcode) have been removed, reads are filtered for their quality in the first pre-processing step. The reads are trimmed to satisfy the desired quality standard and dropped if they do not match quality and/or minimal length requirements [2]. In the next step, overlapping reads can be assembled to longer genomic fragments (contigs) not requiring a reference genome (de novo) [1,2]. Without a reference genome, de novo assembly is computationally challenging, comparable to trying to solve a puzzle without knowing the final picture. The subsequent binning of contigs is then used to group contigs according to their organismic origin. Finally, the gene calling and annotation of binned contigs deliver information on the functional potential of individual species. A glance at the literature reveals that numerous strategies and analysis tools for all these steps are available [1,2]. Several tools have been developed to analyze the taxonomic and, in some cases, the functional repertoire of complex microbial communities. It has also been considered to swap the assembly and binning steps, or to apply them iteratively to ultimately obtain the full genomes of yet uncultured community members as so-called metagenome assembled genomes (MAGs) [5,6].

In the following, we will only focus on de novo assembly approaches based on de Bruijn graphs, instead of overlap-layout-consensus and greedy assembly approaches. In contrast to the other approaches, de Bruijn graph-based methods remain computationally efficient even for the vast amounts of sequence data that have become common [7]. Popular representatives of this approach include ABySS [8], MetaVelvet [9], metaSPAdes [10], MEGAHIT [11], and IDBA-UD [12]. The benefit of de Bruijn graph-based algorithms is the flexibility with respect to the input read length, the efficient error correction, and the ability to handle DNA repeats [1,13]. A key parameter for de novo assembly based on de Bruijn graphs is the *k-mer size*. Prior to assembly, reads are split into smaller fragments of size *k*, defining the nodes in the de Bruijn graph. Directed links connect nodes with overlapping fragments such that longer contigs can be derived via the Eulerian path [1,2,14].

Binning refers to sorting contigs into different bins according to their phylogenetic relatedness [15]. In general, three approaches are used to bin reads: considering sequence composition features such as GC content and tetranucleotide frequencies, considering contig abundance, or a combination of both in the so-called hybrid approach [1,15,16]. The binning tool MaxBin [17], for example, uses tetranucleotide frequencies and scaffold genomic information to generate genomic bins. Finally, genomic features like GC-content, genome completeness and contamination, as well as coverage level and genome sizes, are estimated [17]. Further popular tools for the taxonomic classification of metagenomics data are Kraken [18], Kraken 2 [19], and MetaShot [20].

For the analysis of metagenomic data, automated pipelines are available, including the web-based services MG-RAST [21] and MGnify [22], and the MEGAN6 tool collection [23]. MG-RAST and MEGAN6 are open-source pipelines allowing for the comprehensive analysis and comparison of data sets [23,24,25]. Both pipelines compare sequence data against a reference database using BLAST algorithms and forgo time-consuming and error-prone assembly steps [23,24,25]. Here, no in-depth prior knowledge on the microbial system under study or specific bioinformatics expertise is necessary. For MG-RAST, a metadata file can be provided (specifying information on the project; sample, metagenome library, and sample type, such as biofilm, wastewater, or soil), but other prior knowledge cannot be incorporated into the analysis. Such knowledge exists for well-studied systems such as anaerobic digestion (AD) or the human gut, for which characteristic microbial compositions, or enterotypes, have been reported [26,27]. Such prior knowledge on microbial species expected to be present may be helpful in analysis. For example, it can help to focus analysis on hitherto unknown genomes and to increase the chance of high-quality de novo MAG assembly while at the same time reducing computational time.

In this study, the popular generic automated pipelines MG-RAST and MEGAN6 for taxonomic and functional analysis were compared with a custom-made pipeline (CMP). For setting up this pipeline, different popular assemblers including ABySS [8,28], MetaVelvet [9,29], metaSPAdes [10], MEGAHIT [11], and IDBA-UD [12,30], and the binning tools MaxBin2 [17,31] and MetaBat [32] were applied to simulated mock data sets (MDSs) of five bacterial and archaeal species typical for anaerobic digestion at different abundances per data set. Results were compared with respect to quality characteristics (N50 value, maximum contig length, etc.). Thereafter, the best assembler and binning tool were integrated into the CMP. This pipeline was also validated by using the same three simulated mock data sets as before, but now considering the prior knowledge on typical species being present, taking AD as an exemplary microbial process. The derived community compositions using both automated pipelines and the CMP were compared to the known community composition (ground truth) of the mock data sets. Automated pipelines do not allow for the finetuning of individual analysis settings and parameters, whereas the custom-made pipeline provides such a possibility and gives transparent insights into individual analysis steps, intermediate results, and corresponding statistics. Besides evaluating whether the different tools/pipelines were able to correctly identify the composition of the mock communities, we also evaluated how their predictions regarding the communities’ functional repertoire varied. The advantage of our novel CMP is that by focusing on unexpected species, the time-consuming steps (assembly and binning) are performed only on a subset of all reads, reducing runtime and increasing accuracy.

## 2. Materials and Methods

### 2.1. Simulated Metagenomics Mock Data Sets

Three different simulated metagenomic mock data sets were generated and served as test cases to compare the different analysis strategies. A small set of microorganisms repeatedly detected in the process of AD as an example of a biological process were selected to generate mock data sets. In contrast to actual data sets, we choose low diversity communities, excluding closely related microbial species as a best-case scenario. For each mock data set (MDS), two typical but phylogenetically distant archaeal and bacterial species were chosen (Table 1). *Escherichia coli* was chosen as an additional species as an atypical species whose presence is not expected. Its abundance within the community varied between 11.2% and 44.7%. This was done to investigate the effect of a dominating unexpected species, which reduces the fraction of reads associated with the community of interest. In total, three communities (MDS1–3) consisting of the five species yet different abundance distributions were generated (Table 1). Fully annotated genomes from the NCBI Genome database [33] were used to create the MDSs.

MDSs were generated as Illumina MiSeq platform-based shotgun raw reads with a default mutation-distribution for Illumina sequencing platforms by using Grinder [34]. Whole genomes and additional chromosome sequences of *Methanosaeta concilii* were combined in one reference file, while relative abundances were stored within a separate file. Per MDS, two million paired-end reads with lengths of 150 bp ± 100 bp and 300 million bp in total were generated. Taking 3 million bp as the average genome size and considering relative abundances, the expected coverage ranged from 1.3 for the least abundant species to 68.5 for the most abundant species. After that, paired-end reads were de-multiplexed by using the demuxbyname.sh shell script from BBMap [35] to generate two files that contained, separately, the forward and reverse reads for each MDS, being identical to actual Illumina MiSeq platform output files. These two files served as the input for the comparison of different analysis strategies and assessment of tool performance.

Before analysis, the raw reads of MDSs were trimmed, only keeping reads longer than 30 bases, and low-quality bases and fragments were removed, choosing an average quality score of 30. In the following, these processed reads are denoted as *cleaned reads*. Read trimming and quality filtering were performed using Trimmomatic version 0.36 [36].

### 2.2. Tool Evaluation

In light of the wide choice of tools available for each analysis step for implementation in our CMP, we tested only a few popular representatives. Not aiming at an exhaustive comparison, three assemblers and two binning tools were evaluated and applied to all MDSs (Appendix A). The choice of tools was guided by the practical criteria listed in Table 2, and the final selection of tools to incorporate into the CMP was done by considering the quality characteristics listed in Table 3. Based on the different criteria, IDBA-UD and MaxBin2 were included into the CMP (Figure 1).

The exemplary CMP was run on a Linux Ubuntu operation system with 16 GB RAM and an Intel^®^ Core™ i5-4210U 1.70GHz CPU. DIAMOND and MEGAN6, either stand-alone or as part of the CMP, were run on a cluster system with up to 28 nodes and 100 GB RAM. MG-RAST was used as a web-server system.

The automated pipelines MG-RAST and MEGAN6 were prepared as described in Appendix A and applied to the same three MDSs. The comparison of automated pipelines and the CMP focused on relative abundance values for taxa and functional annotations.

### 2.3. Construction of a Custom-Made Pipeline (CMP)

FastQC [38] was used to assess read quality before and after quality filtering and trimming to choose the optimal parameter settings for this pre-processing step. Quality filtering and read length trimming were done with Trimmomatic version 0.36, using an average quality score of 30 and a minimum read length threshold of 30 bp. For the direct comparison of raw and cleaned reads, MultiQC version 1.4 [39] was used.

To include prior knowledge in the analysis, here in the form of microbial species that are known to be part of the community under study, a reference database that comprises the expected species’ genomes was created. Prior knowledge might be available as the system at hand has been described in detail before or in the form of previous taxonomic analysis based on amplicon sequencing or other fingerprinting approaches. In the absence of such knowledge, the custom-made pipeline can be extended by Metaxa2 [40], which infers species being present in the sample from the metagenomic shotgun data itself. Metaxa2 classifies taxonomies after the identification of rRNA genes [40]. Complete genomes of these species are then downloaded as FASTA files from the NCBI genome database. In our example focusing on AD, our reference database of expected species contained 60 archaeal and bacterial species typically associated with AD (Appendix A,) [16,41,42,43,44].

Pre-mapping refers to the step of creating alignments by using Bowtie2 [45] to separate expected and unexpected species related reads to reduce the data volume for the time-consuming de novo assembly step. Bowtie2 aligns cleaned reads against the database of expected species (Appendix A). Based on this alignment, expected and unexpected reads were separated using SAMtools [46,47] and BAMtools [48], with reads which could not be aligned indicating unexpected species. The selection was done based on SAMtool flags (Appendix A). Next, the transformed FASTQ files based on generated BAM files were transformed into FASTA files by using the FastX-Toolkit [49] or alternatively by fq2fa provided by IDBA-UD [12,30]. Finally, one file containing expected reads and another file containing unexpected reads were created.

Expected reads were annotated with respect to taxonomy and function using DIAMOND. For this purpose, the NCBI NR-database was translated into a DIAMOND database (makedb). The expected reads were then aligned against this database with DIAMOND using the blastx algorithm, generating a diamond alignment archive (daa). In order to make this file readable for MEGAN6, the file was translated into a read-match archive (rma) that combines both taxonomic and functional information. In order to generate the rma file, the following databases were used: the EggNOG database [50] to investigate the functional information [51], and the protein accession for NCBI taxonomy-based mapping [52]. For the annotation process, default parameter settings were applied (Appendix A).

Unexpected reads, i.e., reads that could not be mapped to genomes of the reference database, were used for de novo assembly and binning. Interleaved unexpected reads were used as input for the assembly using IDBA-UD with a k-mer range between 51 and 201 (for parameter details see Appendix A). Only contigs larger than 1000 bp were considered. Assembly results were compared by QUAST [53]. After assembly, obtained contigs were binned by MaxBin2 [17,31], using default parameter values and selecting the marker set 40, which is specific for archaeal and bacterial communities (for parameter details see Appendix A). CheckM [54] was used to analyze the resulting bins. After binning, a second step of assembly (re-assembly) was done per bin, only taking those reads that had been assigned to the bin. The reordering of binned reads based on contig bins was integrated in MaxBin2 by the “reassembly” option.

Finally, the contigs were annotated using MEGAN6 with the same databases as before. The procedure is identical to the one applied to the expected reads, except for using the MEGAN6 option for long reads, which enables the discovery of more than one gene per read.

All previous analysis steps result in two annotation files. One file contains annotated expected reads and the other, annotated unexpected contigs. In order to combine both files, contigs must be decomposed into their constituting reads. For this purpose, Bowtie2 was used to map unexpected reads to contigs. Finally, counts and annotation of expected and unexpected reads were summarized (Appendix A). Ultimately, absolute read counts were normalized based on the total number of reads per sample to generate relative abundances of community composition and functional potential for comparison across samples.

In order to analyze taxonomic and functional information in combination, the taxonomic and functional annotation files containing count data generated by MEGAN6 were combined into a single taxonomy vs. function table using a custom R script (Appendix A).

The whole workflow of the CMP is illustrated in Figure 1, and all used commands are summarized in Appendix A. The CMP can be downloaded at https://git.ufz.de/UMBSysBio/cmp/.

## 3. Results

### 3.1. Tool Selection

Five assemblers were compared with respect to N50 value, largest contig size, total assembly length, and number of contigs. When comparing assembly tools using two k-mer length settings for MetaVelvet and ABySS (*k* = 55 and *k* = 83), and setting the k-mer length range to 51–201 for IDBA-UD, metaSPAdes, and MEGAHIT, ABySS delivered the largest number of contigs (14,267 for *k* = 83) and produced the largest contig size (421,238 bp for *k* = 55, Appendix A). However, the results differed considerably when changing the k-mer length. MetaVelvet delivered the lowest N50 value (2341 bp). IDBA-UD, metaSPAdes, and MEGAHIT delivered comparable results regarding the largest contig size and total assembly length. With respect to the N50 value, metaSPAdes outperformed all the other tools (reaching up to 66,131 bp) and paired with the lowest number of contigs, produced the most contiguous assembly, followed by IDBA-UD.

Regarding binning, the tested tools MaxBin2 and MetaBat generated the same number of bins per MDS (2–3 bins) and a similar taxonomic marker-based assignment (Appendix A). As the downstream processing requires binned reads instead of binned contigs, we adopted MaxBin2 as the binning tool for our CMP as it supports this option. As IDBA-UD is included in MaxBin2, we chose IDBA-UD over metaSPAdes as the default assembler for the sake of simplicity of installation.

### 3.2. Application of Custom-Made Pipeline to Mock Data Sets

The CMP approach was evaluated using three mock data sets (MDSs), each containing two million paired-end reads. Pre-processing (quality filtering and read length trimming) removed less than 0.3% of all input reads. In the next step, reads were mapped against an AD-specific genomic database of expected species (Appendix A). The overall mapping rates for the cleaned reads of MDS1, MDS2, and MDS3 were around 32%, 61%, and 75%, respectively. These results correctly reflected the decreasing share of *E. coli* in the mock community as an unexpected species. Unexpected reads, not mapping to the AD-specific database, were then assembled de novo with IDBA-UD. Thereby, MDS1 produced the largest contig with 327,353 bp (Table 1) as well as the largest N50 value (133,300 bp). The total assembly length of MDS1 and MDS2 was around 4.5 million bp, whereas MDS3 generated a total assembly length of around 3 million bp. At most, two bins could be generated from the unexpected reads of our MDSs. MDS1 generated only one bin, which contained all contigs equal to the assembly (noclass-bin). Reads contributing to contigs belonging to one bin were separately assembled again per bin (reassembly). Here, the total assembly length of MDS3 increased from around 3.0 Mb to 3.4 Mb. The largest contig size and N50 values were equal to those of the initial assembly or increased after reassembly (Table 4 and Appendix A).

### 3.3. Comparison of Community Composition

We compared automated pipelines with our CMP, which allows for the inclusion of prior knowledge with respect to species expected to be present in the microbiome of interest (Methods for more details). To compare the pipelines, we analyzed the three mock communities separately with MEGAN6, MG-RAST, and our CMP and compared the results on different taxonomic levels (Appendix A) with the actual community composition.

Overall, both the CMP and MEGAN6 showed differences between predicted and actual community composition (Figure 2a). While the CMP underestimated *Methanosaeta*, MEGAN6 overestimated *Syntrophomonas* and *Methanoculleus* (Figure 2b). If *Methanosaeta* is the most abundant species of the actual community composition (>50% relative abundance), then *Methanosaeta* was less underrepresented during CMP analysis. The higher the percentages of unexpected species (here *E. coli*), the more overrepresented *Escherichia* was in the CMP (up to 14% when the actual share was about 11%), and it was underrepresented down to 7% using MEGAN6 only. Community compositions predicted by MG-RAST deviated strongly from actual mock community compositions. Although the trend of smaller *Escherichia* fractions in the community from MDS1 to MDS3 was correctly predicted, *Escherichia* was predicted to dominate the community in all cases (with at least 59%). The proportions of unassigned reads were on average only 1.4% for CMP and 2.1% for MEGAN6. The number of unassigned reads decreased as the number of expected reads increased over the MDSs for CMP and MEGAN6. MG-RAST assigned all reads.

The inferred richness on the genus level varied depending on the pipeline. MEGAN6 (stand-alone) detected in total up to 11, CMP up to 19, and MG-RAST more than 699 different genera. Focusing only on dominant species with a relative abundance above 1%, the richness according to MG-RAST analysis decreased strongly to, on average, five dominant genera. This was in the same range as the results from the CMP and MEGAN6 with seven genera having a higher relative abundance than 1%. For more detailed relative abundance information on the expected and unexpected species, Appendix A.

On the highest taxonomy level, the CMP and MG-RAST were able to reproduce the increasing share of archaea from MDS1 to MDS3. MG-RAST, however, strongly overestimated the bacterial share, and MEGAN6 wrongly predicted a falling share of archaea (Appendix A).

### 3.4. Comparison of Functional Potential

Besides comparing inferred community compositions, we also used the three analysis strategies to infer the functional potential of the three mock communities. The functional annotation in CMP and MEGAN6 was based on the EggNOG database, while the COG database was used for MG-RAST, for which EggNOG annotation was not available. In general, the differences in the functional potential between the analysis pipelines were much smaller than for the compositional analysis (Figure 3). For EggNOG level 1 and COG classification, the largest fraction of reads was assigned to metabolism (around 35% overall), followed by information storage and processing (around 17% overall), and cellular processes and signaling (around 17% overall). This pattern was similar across mock communities and analysis strategies. One notable difference was, however, the number of unassigned reads. MEGAN6 featured the most unassigned read counts (41–51%), followed by CMP (24–30%), and MG-RAST (19–20%) (Appendix A). 

While MEGAN6 allows for the extraction of all read/contig annotation information in text files (read counts per taxa or functional level, taxa or functional name, taxa or functional path, etc.), this was not possible with MG-RAST. Our CMP uses DIAMOND [55] and MEGAN6 for taxonomic and functional annotation, and hence allows for the combination of both. Taking the functional classification on level 1 using the EggNOG database per species and sample, we find that relative species- and function-specific counts agree well with the shifting community compositions from MDS1–3, containing less *Escherichia* and more *Methanosaeta* reads (Figure 4).

### 3.5. Advantage of Including Expected Species as Prior Knowledge in Analysis

Besides the ability to correctly predict community composition, the required computational time is another important factor for the analysis. In general, the computational time varies greatly between MG-RAST, MEGAN6, and CMP. MG-RAST required between 1 week and 3 months for three MDSs using the low priority option, MEGAN6 with prior BLASTX alignment (using DIAMOND) required 4.5 days on a cluster system, and the CMP ran for around 3 days, or 1.5 days if using the prior knowledge option, on a standard desktop PC (the annotation was done on the same cluster system as when using MEGAN6 as a stand-alone tool).

To assess the advantage of CMP’s ability to incorporate prior knowledge into analysis, we compared CMP results when using the pre-mapping step to separate expected from unexpected reads and those when skipping this step. In the second case, the total interleaved reads were assembled and binned using tools as indicated in the Methods section. Then, both types of bins (only unexpected read bins and total paired-end read bins) were annotated per bin using DIAMOND and MEGAN6 using the NCBI-NR database for taxonomic classification. Table 2 illustrates the results of taxonomic classification per bin (for details see Appendix A).

The N50 value of the assembly of unexpected reads (on average 85,000) when using the pre-mapping option of the CMP was higher than the N50 value of total cleaned interleaved reads (average 57,000) (more details under Appendix A). As expected, the CMP with pre-mapping produced fewer bins than without pre-mapping as only *E. coli* based reads were expected to not be filtered by the pre-mapping step as *E. coli* was not contained in the database of expected species. Besides *E. coli*, up to 12% of reads were identified as other species (Table 5). But all these species were part of the family *Enterobacteriaceae*, and consequently, these species were related to *E. coli.* Thus, nearly perfect bins were generated. The CMP without the pre-mapping step shows a higher completeness (up to 99%) but also a higher contamination rate (up to 4%) compared to the CMP with pre-mapping, which shows a maximal completeness of 53% and a contamination rate of up to 0.6%. The few unassigned reads (<2%) are mainly short reads, sequencing artefacts, or reads of unknown species. The annotation of bins based on total interleaved reads showed more bins per MDS. In the best case, five bins should be generated, matching the number of species present in the community. However, fewer bins per MDS were generated and the content of bins showed more than one dominant species. Bins were most often dominated by *M. concilii* with 46–100% relative abundance, but these bins additionally contained many reads of other species. Particularly little abundant species like *A. colombiense* were clustered in one of the other bins.

## 4. Discussion

A whole plethora of tools has become available for the analysis of metagenomic data. While automated analysis pipelines are easy to use, they lack flexibility and the ability to incorporate prior knowledge that might be available when analyzing systems like anaerobic digestion processes. Here, we show, based on mock data sets of low-diversity communities, that the inclusion of knowledge about expected species in our custom-made pipeline can improve the quality of bins and reduce the required runtime in comparison to popular automated pipelines. The benefits of focusing time-consuming steps on unexpected species rather than on all species are likely to be even more pronounced for natural communities of higher diversity. For the community composition, the CMP and MEGAN6 generated similar results. The assembly-free analysis strategies of MEGAN6 and MG-RAST can be more sensitive, because a greater proportion of species inside the annotated reference genome or functional database is integrated into the analysis process, but less specific in functional identification, as the functional results showed [56]. MG-RAST successfully matched all reads against a database, resulting in a higher richness of species than present in the initial community. If only species having a relative abundance larger than 1% are considered, richness decreased strongly and was more comparable with the results of MEGAN6 and the CMP.

CMPs provide several advantages in comparison to automated analysis pipelines. Firstly, regardless of the analysis goal, CMPs enable the arrangement of individual tools in different workflows (flexibility), for example, allowing for an easy reordering of binning and assembly steps. Secondly, the outputs of all steps are recorded and can easily be inspected as an important step in troubleshooting (transparence). Thirdly, the integration of prior knowledge leads to smaller data sets for the time-consuming steps like assembly, binning, and annotation. This was achieved in our case by excluding reads originating from species that were expected to be present in our samples and for which genomes are available. This not only leads to faster computation times but also lowers the contamination rate during the binning step, which simplifies the recovery of novel genomes as metagenome-assembled genomes as a possible next analysis step. The required computational time varies greatly between the CMP, MG-RAST, and MEGAN6. The benefits with respect to computational time reported for the CMP must, however, be seen in the light of the extra time required for pipeline preparation and implementation. However, once it is prepared, the analysis of data from further samples is fast, and there are many possibilities to adapt the pipeline to changing needs with minimal effort.

The modularity of the CMP also allows for an easy incorporation of updates of individual tools. For example, an alternative pre-processing tool is CUTADAPT [57], which provides a similar functionality to Trimmomatic [36]. As an additional alternative to Bowtie2 [45], the aligner BWA [58] works well. Bowtie2 is the typical choice to generate alignments in combination with the SAMtool [46,47] modification functions. Bowtie2 is reliably applied in numerous metagenomics-based studies and also useable for Illumina-based reads of different lengths [45,59]. It works well, with a high alignment rate for reads that contain errors, and provides better accuracy [59]. A BLASTX-based approach to map translated DNA reads against a target protein sequence by using the EggNOG database was chosen to annotate the functional information with MEGAN6. Alternatively, a hidden-Markov model (HMM) approach could be used. However, DIAMOND is faster during searching the best orthology of every query against all EggNOG proteins than HMM and is thus more recommendable for large data sets, which are common in metagenomics analyses [60,61].

The pre-mapping step of our CMP is best applied to microbial systems for which enterotypes or typical microbiomes have been reported, such as in AD. An exemplary database suitable for the pre-mapping step for AD microbiomes is listed under Appendix A. The CMP is, however, also applicable to other systems such as gut systems, activated sludge, or marine microbiomes. This requires the compilation of a suitable species database of expected species for the pre-mapping step. The size and contents of the database have a profound impact on the results. If it contains the most abundant species in the community, only a few reads will remain for the de novo assembly of unexpected species. Nevertheless, if sequencing depth is deep enough, the pre-mapping step will increase the chance of the successful recovery of MAGs of low-abundance species. If the database does not contain the most abundant species in the community, for example, if dealing with an uncommon representative of the respective process, the benefit of the pre-mapping step is severely diminished. In general, if more species are included in the database and these species are indeed in the community, this leads to relatively fewer reads becoming available for de novo analysis and bins of higher quality, at the expense of computational time. As the database size increases, and if it also contains non-present species, incorrect read alignments become more likely. As both the selection of the species and the actual abundance distribution of the community affect the results, it is difficult to formulate a general rule for creating the optimal database for the pre-mapping step. To enhance phylogenetic resolution, one can also include pan genomes in the database, which, for example, can be created by Roary [62]. This requires, however, that genome sequence information for sufficient numbers of species of interest are retrievable from sequence databases, which might not be the case for underexplored microbial systems. As for other approaches, the presence of closely related species in the community reduces the chance of MAG reconstruction for one of these species. Let us assume that two species share 80% of their genome, and for only one of them, the genome is available and contained in the database of expected species. If both species are present in a community, reads of the novel species associated with the 80% of identical sequences will be recruited to the reference database and only reads associated with the 20% of unique sequences will end up as unexpected reads. This will not enable the MAG reconstruction of the novel species. This situation, however, can be detected in principle by investigating the coverage along the genome contained in the database. In this example, two distinctive coverage levels would be observed, a higher coverage along the 80% of shared sequence, and a lower coverage along the unique sequences. Such an observation would flag this situation, provide an estimate of the amount of shared sequences, and might still enable MAG reconstruction by combining unexpected reads with an appropriate selection of reads recruited to the reference genome in question.

While creating the database requires more in-depth bioinformatics knowledge than using web-based automated pipelines, it does not require the level of dexterity required for setting up a full pipeline from scratch. While the selection of appropriate species for the database requires expert knowledge on the process under investigation, a lack of this knowledge can be compensated for to some extent by using tools like Metaxa2, which infer community composition by identifying rRNA marker genes in metagenomic data [40]. This information can serve as a basis to construct an appropriate database of expected species.

## 5. Conclusions

Acquiring meta-omics data on complex microbial systems has become a standard procedure over recent years. Advances in experimental approaches and in bioinformatic analysis have enabled deep insights into these systems. The data analysis step should not be underestimated as it has a big impact on the obtained results with respect to community composition and functional potential. Currently, one can choose between easy-to-use automated pipelines or the construction of application-tailored pipelines combining existing tools. During the selection process for the analysis strategy and pipeline, user friendliness and computational runtime performance should be assessed. We here showed that applying a custom-made pipeline might be worth the extra effort in comparison to using existing pipelines as it offers benefits in terms of flexibility, transparency, and results. In particular, our pipeline allows for the inclusion of prior knowledge in terms of expected microbial species, leading to faster runtimes and a better characterization of the yet-unknown part of the microbial system under study.

## Figures and Tables

**Figure 1 microorganisms-08-00669-f001:**
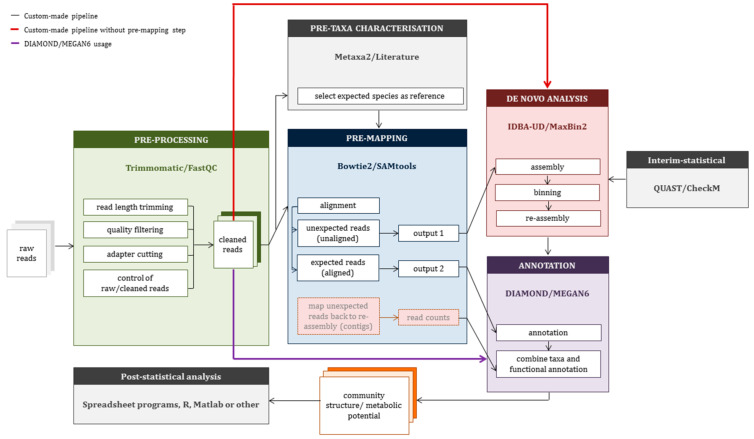
Metagenomic analysis by using MEGAN6 stand-alone or a custom-made pipeline approach. Custom-made pipeline to analyze metagenomic data by using prior information on expected microbial species (black arrows). Alternatively, the pre-mapping step can be skipped and de novo assembly (red arrow) or annotation by using MEGAN6 (purple arrow) be done directly using cleaned reads.

**Figure 2 microorganisms-08-00669-f002:**
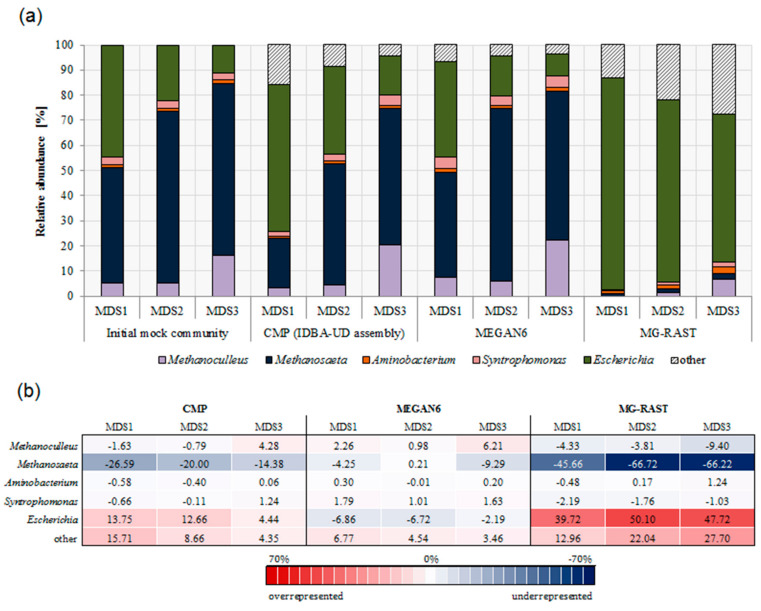
Comparative community analysis. (**a**) Comparison of community compositions on the genus level predicted by different analysis strategies based on three mock communities, MDS1–3. The custom-made pipeline (CMP) integrates prior knowledge on the community and includes an assembly step, while MEGAN6 and MG-RAST represent two automated analysis pipelines without assembly; (**b**) Deviation of predictions from actual species fractions in the total community. The section “other” refers to identified genera that were not part of the initial community composition and unassigned reads.

**Figure 3 microorganisms-08-00669-f003:**
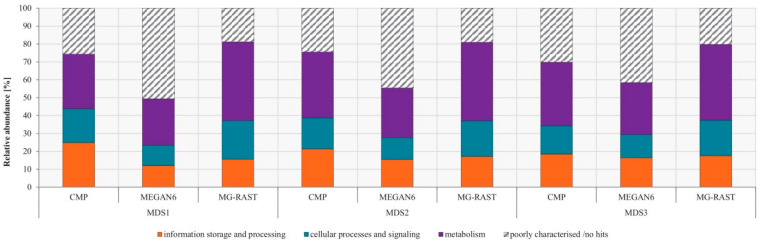
Functional potential of mock communities compared between analysis strategies. The relative abundances were calculated by using absolute read counts per sample and analysis approaches. Annotation was based on the EggNOG (CMP and MEGAN6) and COG databases (MG-RAST).

**Figure 4 microorganisms-08-00669-f004:**
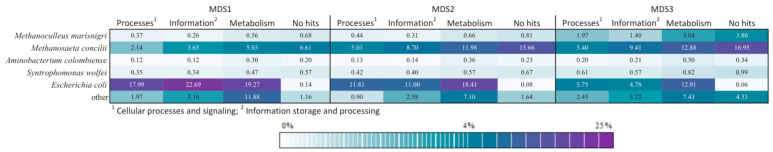
Functional potential combined with taxonomical information using the custom-made pipeline. Counted reads per sample were used to calculate the relative abundance. The functional potential analysis uses the EggNOG database, level 1 classification. The row “other” refers to identified species that are not part of the initial community composition and unassigned reads (for detailed relative abundances see Appendix A).

**Table 1 microorganisms-08-00669-t001:** Composition of metagenomic mock data sets.

Species	NCBI Reference Sequence	Genome Size [bp]	Abundance [%]		
			MDS1	MDS2	MDS3
*Methanoculleus marisnigri*	NC_009051.1	2,478,101	5.03	5.08	16.22
*Methanosaeta concilii* ^1^	NC_015416.1 NC_015430.1	3,008,626 18,019	46.08	68.45	68.44
*Aminobacterium colombiense*	NC_014011.1	1,980,592	1.35	1.35	1.34
*Syntrophomonas wolfei*	NC_008346.1	2,936,195	2.80	2.80	2.80
*Escherichia coli*	NC_000913.3	4,641,652	44.74	22.37	11.19

^1^ Combination of complete genome (NC_015416.1) and a selected chromosome sequence (NC_015430.1).

**Table 2 microorganisms-08-00669-t002:** Criteria to select tools for metagenomics data analysis [37].

Criteria	Description
Availability	Open source or commercial version as download or web service.
Support	Availability of: manual/readme, version update, help functions, and/or developer contact.
Flexibility	Flexible usage (variety of input/output data formats).
Run time	Expenditure of time: Operating time.
Usability	Ease of tool usage (installation, parameter settings, algorithm applicability).

**Table 3 microorganisms-08-00669-t003:** Quality characteristics to compare tool output.

Step	Feature	Description
Assembly	Number of contigs	Number of contigs generated
	Largest contig	Length of the longest contig
	N50	Median statistic whereby 50% of the entire assembly (in terms of number of bases) is contained in contigs longer or equal to the N50 value. Assemblies can only be compared if the assembly size is similar [14].
	Total assembly length	Sum over all bases of all contigs generated
Binning	Number of bins	Number of bins (clusters)
	Completeness/contamination	CheckM compares completeness and contamination per bin depending on a defined number of marker genes for bacterial and archaeal genomes as a quality factor.

**Table 4 microorganisms-08-00669-t004:** Comparing assembly and reassembly results.

Criteria	MDS1	MDS2	MDS3
**Assembly**			
number contigs	66	167	966
largest contig [bp]	327,353	145,780	16,676
total assembly length [bp]	4,564,668	4,520,232	2,975,790
N50 [bp]	133,300	47,255	3,611
**Reassembly**			
number contigs	72	75	672
largest contig [bp]	327,353	317,670	27,558
total assembly length [bp]	4,567,420	4,378,346	3,381,373
N50 [bp]	132,848	113,839	5,862

**Table 5 microorganisms-08-00669-t005:** Composition of bins compared between the CMP with and without pre-mapping for all three mock data sets (MDS) in relative abundances (%). Pre-mapping successfully removes reads associated with expected species, only leaving *E. coli* reads for assembly and binning.

		CMP with Pre-Mapping (Only Unexpected Reads)						CMP without Pre-Mapping(All Reads)					
MDS	Bins	MM	MC	AC	SW	EC	other	MM	MC	AC	SW	EC	other
1	nc	0.00	0.00	0.00	0.00	90.30	9.70	16.25	3.48	19.33	59.35	0.20	1.39
	1	--	--	--	--	--	--	0.00	2.22	0.00	0.00	88.25	9.53
	2	--	--	--	--	--	--	0.00	99.60	0.00	0.27	0.00	0.13
2	nc	0.00	0.00	0.00	0.00	89.80	10.20	13.73	14.89	15.81	49.17	4.76	1.64
	1	0.00	0.00	0.00	0.00	96.63	3.37	0.65	81.42	0.79	2.55	13.33	1.26
	2	--	--	--	--	--	--	0.79	50.57	0.95	3.10	40.27	4.32
3	nc	0.00	0.00	0.00	0.00	89.21	10.79	0.96	6.52	13.19	39.82	35.36	4.16
	1	0.00	0.00	0.00	0.00	94.38	5.62	3.26	82.57	1.05	3.47	8.85	0.80
	2	0.00	0.00	0.00	0.00	88.12	11.88	18.55	59.36	1.50	4.93	12.67	2.99
	3	--	--	--	--	--	--	3.46	45.54	1.15	3.77	41.53	4.55

Unit = % | nc = noclass (contigs which could not be assigned to a bin) | MM = *Methanoculleus marisnigri |* MC = *Methanosaeta concilii |* AC = *Aminobacterium colombiense |* SW = *Syntrophomonas wolfei |* EC = *Escherichia coli |* other = all species beside the initial community are summarized | Bold numbers highlight the highest relative abundance per bin. | -- = No bins

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
