# Peer review of "A Modular Metagenomics Pipeline Allowing for the Inclusion of Prior Knowledge Using the Example of Anaerobic Digestion"

_microorganisms, 2020, doi:10.3390/microorganisms8050669_

Round 1

Reviewer 1 Report

As attached file

Author Response

Table 5: Is difficult to interpret, I would expect to see similar results with and without pre-mapping and levels across the 5 species used with, in an ideal world, the pre- mapping option providing better results and decreased computational time, please increase the explanation of the table either in the text or the legend.

Legend states: Bold numbers highlighted the highest relative abundance per bin, however, there are no bold numbers.
It is not clear to what the MDS rows are referring in the legend, are these replicates/bins?

CMP with pre-mapping only shows E.coli?? CMP without pre-mapping shows little E.coli – why were only E.coli based reads expected to not be filtered by pre-mapping (lines 359, 360)?? Did the initial dataset not include all 5 species?

Response: We improved the table (formatting, labeling, caption, and applying bold font for peak entries, as promised in the caption). Rows MDS1-3 refer to the three mock data sets, for which 2 or 3 bins were generated. We clarified in the text and in the caption, that the pre-mapping step removes all reads associated with expected species contained in the database, only leaving unexpected species (here: E. coli) for the assembly and binning step. (Table 5, P. 11, L. 384-385; locations refer to the track-change version of the revised manuscript).

Minors:

Line 56/57: We will only focus on de novo assembly approaches based on de bruijn graphs...
It would be good here to include a sentence to explain why not including overlap- layout-consensus methods and greedy assembly approaches

Response: We have added a note on the superior computational efficiency for large sequence data sets of de Bruijn graph based methods (P. 2, L. 58-60).

Line 114: the community was varied between [delete was]

Response: We corrected the sentence (P. 3, L. 123-124).

Line 185: ‘unexpected’ and ‘expected’ reads – it is not explained until line 200 that these are unaligned and aligned reads to the refence genomes? If so it would be clearer to state that at the beginning of this section.

Response: Yes, this is the idea; we added a clarifying statement (P. 6, L. 205-206).

Section 3.1 Tool selection – is there any reason why not to use SPAdes assembler with metaSPAdes – this is a very popular tool that people may be looking out for?

Response: When starting the project, metaSPAdes was not yet popular enough to be considered; we now added metaSPAdes and also MEGAHIT in our assembly comparison, both yielding comparable results to IDB-UD. (Additional File 1, Figure A.1, Section A2, additional References in the main text, P. 2, L. 61, L. 91-92).  

Line 412: Additional, besides Bowtie2... should read As an additional alternative to Bowtie2,

Response: We corrected the sentence (P. 12, L. 441).

Discussion: The discussion covers the points that come up in the text, specifically for this reviewer that the choice of species for the pre-mapping step requires expert knowledge and that including species of interest but without specific pathways of interest (not present in the chosen reference isolate of a particular species but present in the dataset). Indeed including pangenomes instead of a specific isolate would work, however, many of the species of interest are underrepresented in the sequence databases thus this may not be possible.

Response: We added a qualifying statement regarding the reliance of our approach on the contents of sequence databases (P. 13, L. 471-473).

Reviewer 2 Report

The manuscript by Daniela Becker et al, interestingly introduce a new pipeline designed for the analysis of metagenomic data allowing to guide the analysis by using prior knowledge. Nonetheless, according to the shown results I have several concerns about the proposed approach.

Introduction:

Line 39 – 40: a more general definition should be tools performing the direct taxonomic bin of reads. Moreover the list of this tools should be increased by adding tools Kraken, Kraken2 , MetaShot or mOTU2.

Line 58-59 regarding metagenome assembly, tools like metaSPAdes and megahit should be at least cited.

Line 78-80 What should contain the metadata file supplied to MG-RAST? Just information associated to the samples or something else. Please specify.

Line 87-91 I cannot understand the choice of the assembly tools: ABySS was not designed for Metagenome analysis, MetaVelvet is a re-implementation of the Velvet assembler and actually works on a single kmer length and the usage of multiple kmer lengths is quite difficult. Only IBDA-UD seems to correctly address the task. I would like to know the reason why authors decided do not test tools like metaSPAdes or megahit.

Materials and Methods:

Line 116-118: the MDSs do not resemble a real metagenomic dataset: the complexity is generally really low compared to real dataset. Moreover, no species belonging to the same genus were included.

Line 125-126 the number of generated reads do not resemble a real dataset even considering only the 5 species included in the MDSs. Assuming a total genome size is 15 Mb (the sum of all genome size), the expected coverage is around 20. Moreover, considering the used genome widely variates in genome size (from 1.9Mb to 6.4 Mb) an higher coverage may help the assembly.

Table1: Add the genome size

Line 138-140 I agree with the authors that the aim of the manuscript does not regard the comparison of tools but again I suggest to include at least metaSPAdes.

Additional file section A.1 Havethe authors also tested shorter kmers? In order to improve the connectivity the usage of short kmer (29,31,33) may improve the assembly contiguity

Line 192-180 Here resides my major concerns. Let’s suppose the metagenome under investigation contains two phylogenetically closed genomes (i.e. strains of the same species) differing for less than 20% of their genome and in the expected list just one of this is listed. I expect following the bowtie mapping the portion of reads put into the unexpected file for the unlisted strain may be not sufficient for an efficient assembly. More generally, if the expected list contains genomes belonging to the same family or order, the pre-mapping may produce a loss of information due to genomic similarities between the bacterial species. Just mapping reads on the “expected” genomes without any other check may produce a loss of informations.

Line 194 correct daa to data

Results

Line 232-245 The obtained results may suffer the imposed kmer length: shortening the kmer length should improve the connectivity.

Line  260 – 262 Do you mean Mb?

Line 272-275. The obtained result may converge to my concerns. In the expected list there are 4 Methanosaeta genome (one in the MDS and 3 not) and other genomes belonging to the same order. The reduction of Methanosaeta underrepresentation may be explained by the fact that their read maps also to genomes belonging to non-co-generic species. Probably the pre-filtering step should be tuned by considering other parameters.

Author Response

Line 39 – 40: a more general definition should be tools performing the direct taxonomic bin of reads. Moreover the list of this tools should be increased by adding tools Kraken, Kraken2 , MetaShot or mOTU2.

Response: We added this more general definition and the mentioned tools, except mOTU2, as this is based on a selection of marker genes (P. 1, L. 39, P. 2, L. 73-74; locations refer to the track-change version of the revised manuscript).

Line 58-59 regarding metagenome assembly, tools like metaSPAdes and megahit should be at least cited.

Response: When starting the project, metaSPAdes was not yet popular enough to be considered; we now added metaSPAdes and also MEGAHIT in our assembly comparison, both yielding comparable results to IDB-UD. (Additional File 1, Figure A.1, Section A2, References in the main text: P. 2, L. 61, P. 2, L. 91-92).

Line 78-80 What should contain the metadata file supplied to MG-RAST? Just information associated to the samples or something else. Please specify.

Response: We added a clarification (P. 2, L. 81-82).

Line 87-91 I cannot understand the choice of the assembly tools: ABySS was not designed for Metagenome analysis, MetaVelvet is a re-implementation of the Velvet assembler and actually works on a single kmer length and the usage of multiple kmer lengths is quite difficult. Only IBDA-UD seems to correctly address the task. I would like to know the reason why authors decided do not test tools like metaSPAdes or megahit.

Response: ABySS has been used for metagenomics data before (e.g., Oulas et al., 2015, doi: 10.4137/BBI.S12462; or Ju & Zhang, 2015, doi: 10.1021/acs.est.5b03719). Indeed, the choice of the k-mer length is crucial and it is common practice to test a range of k-mer lengths to obtain optimal results. While IDBA-UD (also metaSPAdes, MEGAHIT) internally tests a range of k-mer lengths (options –mink, --maxk), for other assemblers (such as MetaVelvet) this can be easily emulated by calling the assembler in a scripted loop over different k-mer length values, which we also implemented. metaSPAdes and MEGAHIT are now added to the comparison (see above).

Materials and Methods:

Line 116-118: the MDSs do not resemble a real metagenomic dataset: the complexity is generally really low compared to real dataset. Moreover, no species belonging to the same genus were included.

Response: We chose an idealized “best-case” scenario for our test, which we now acknowledge explicitely (P. 3, L. 119-121). Indeed, for a full evaluation which is not intended in this work, a range of low to high diverse communities containing both phylogenetically close and distant species would be required.

Line 125-126 the number of generated reads do not resemble a real dataset even considering only the 5 species included in the MDSs. Assuming a total genome size is 15 Mb (the sum of all genome size), the expected coverage is around 20. Moreover, considering the used genome widely variates in genome size (from 1.9Mb to 6.4 Mb) an higher coverage may help the assembly.

Response: We added the information that if considering the chosen relative abundancies, the coverage varied between 1.3 and 68.5 for the respective species (P. 3, L. 136-138).

Table1: Add the genome size

Response: We added the information (Table 1).

Line 138-140 I agree with the authors that the aim of the manuscript does not regard the comparison of tools but again I suggest to include at least metaSPAdes.

Response: We included metaSPAdes and MEGAHIT in the comparison (see above).

Additional file section A.1 Havethe authors also tested shorter kmers? In order to improve the connectivity the usage of short kmer (29,31,33) may improve the assembly contiguity

Response: We did not tested such small values but followed multiple advice recommending k-mer length >= 51 for reads longer the 65bps (MetaVelvet), and k-mer length recommended to be in the range of the read length (Chikhi & Medvedec, 2014; doi: 10.1093/bioinformatics/btt310). When testing all k-mer lengths between these extremes on an actual data set using reads of 251 bp length (as in our mock data set), we observed degrading N50 and contig length values towards the short end, hence not suggesting a better performance for shorter k-mer lengths.

Line 192-180 Here resides my major concerns. Let’s suppose the metagenome under investigation contains two phylogenetically closed genomes (i.e. strains of the same species) differing for less than 20% of their genome and in the expected list just one of this is listed. I expect following the bowtie mapping the portion of reads put into the unexpected file for the unlisted strain may be not sufficient for an efficient assembly. More generally, if the expected list contains genomes belonging to the same family or order, the pre-mapping may produce a loss of information due to genomic similarities between the bacterial species. Just mapping reads on the “expected” genomes without any other check may produce a loss of informations.

Response: This is a valid point which we now discuss in more detail in the discussion. Indeed, our pipeline might fail to allow for an assembly of closely related species due to the mentioned scenario. Such a situation can, however, be detected post hoc by inspecting the coverage along the genome of the species contained in the database: a strongly varying coverage along the genome would indicate, that reads of multiple species were recruited along some parts of the genome (P. 13, L. 473-485)

Line 194 correct daa to data

Response: We modified the sentence to avoid confusion: “daa” refers to a “diamond alignment archive” (P. 6, L. 213).

Results

Line 232-245 The obtained results may suffer the imposed kmer length: shortening the kmer length should improve the connectivity.

Response: See discussion above.

Line  260 – 262 Do you mean Mb?

Response: We corrected this error (P. 8, L. 281).

Line 272-275. The obtained result may converge to my concerns. In the expected list there are 4 Methanosaeta genome (one in the MDS and 3 not) and other genomes belonging to the same order. The reduction of Methanosaeta underrepresentation may be explained by the fact that their read maps also to genomes belonging to non-co-generic species. Probably the pre-filtering step should be tuned by considering other parameters.

Response: The computation of relative species abundancies in the community is actually independent of the filtering process, as both filtered and unfiltered reads will end up in the final annotation step where they are jointly evaluated (Figure 1).

Round 2

Reviewer 2 Report

The authors have overall improved the manuscript quality.

The Section Result 3.1 needs an improvement by adding the discussion of all the tested assembler. In particular, the tested tools are five and not three.

Moreover, by observing the figura A.1 in Supplmentary materials, it is possible to note how metaSPAdes outperforms the other tools in terms of both N50 and the produced number of contigs, indicating a more contiguous assembly. Considering the other used parameters for the comparison, metaSPAdes results are comparable to IDBA-UD.

Taking into account this it is not clear to me the reason why the authors decided to include IDBA-UD in their pipeline. Authors should clarify it for example by indicating if the choice was guided also by computational/coding requirements.

Author Response

We are grateful for pointing out our oversight: We updated the section accordingly (P. 7, L. 233-242). Also, we now mentioned that IDBA-UD was used due to its ease of installation: it comes bundled with MaxBin2 (P. 7, L. 246-248).

This manuscript is a resubmission of an earlier submission. The following is a list of the peer review reports and author responses from that submission.

Round 1

Reviewer 1 Report

In this study, the authors have presented an approach for the analysis of shotgun metagenomic data. Specifically, their approach attempts to make assembly based analysis of the microbiome less intensive in terms of computation by only assembling reads from unclassified species. Overall, I found the approach to be an intriguing one and I believe that it may be of interest to the wider community. However, I have several concerns about the study.

Major comments:

I feel that there is room for improvement in the proposed pipeline, in these two areas:

Firstly, while I agree that it is valuable to have some awareness of the sort of species which are present in a community, I do not feel that it is practical to rely on literature to construct a database of expected species, especially for environments which are more complex than the ones analysed here. Instead, the alternative of using a classifier such as Metataxa2, as the authors have done, is a preferable approach. It is then possible to programmatically download all of the reference genomes for each of the species that were detected by the classifier. Following on from this, instead of using a single genome from each expected species, the authors might consider using pangenomes (which can be built using a tool such as Roary) from those species. I believe that HUMAnN2 (which relies on the alignment of reads against the ChocoPhlAn database) also uses a pangenome based approach for characterisation of shotgun metagenomics data. Secondly, I am confused about why the authors chose to annotate reads by mapping them against the nr database using DIAMOND, as this is intensive. Since the aim of the pipeline is to reduce the cost of analysis, I believe that it is worth looking at an alternative for annotation, especially if the approach is to be adopted by researchers who do not have access to a HPC. For example, could the authors annotate contigs using a tool such as Prokka? Reads could then be mapped against the contigs to estimate abundances.

Additionally, the authors included a limited number of tools in their comparisons, and additional tools could be included to expand upon the results:

Only 3 assemblers (ABySS, IDBA-UD, and MetaVelvet) were compared Consider including MEGAHIT, MetaSPAdes, etc. Only two binners (MetaBAT and MaxBin2) were compared Consider including CONCOCT, MyCC, etc. Only two automated pipelines (MEGAN6 and MG-RAST) were used Consider using MGnify

It would also have been good to look at more than 3 samples, especially given that the authors used simulated communities. Had the authors made more mock communities, they would have been able to do statistical comparisons between the approaches, which would have been useful. At present, it Figures 2 & 3 appear to show that CMP is similar to MEGAN6, so it is difficult to see if CMP is more accurate than automated pipelines.

Minor Comments:

L15: Replace “to facility this” with “to facilitate this”

L38-51: This section describes a typical metagenome assembly based workflow. The authors should specify this. Also, it would be worth mentioning alternative approaches which rely on short read alignment (MetaPhlAn2, HUMAnN2, etc.).

L43-44: Explain why de novo assembly is challenging

L46-47: Provide references for “A glance at the literature...”

L52-58: Give examples of these assemblers

L60: “sequence characteristics” is vague

L65-66: MGnify is an automated pipeline provided by EBI

L100: Replace “exemplary biological process” with “an example of a biological process”

L411: Replace “over the last years” with “over recent years”

Figures 3 & 4: Reorder the bars so that each approach is side-by-side (e.g. for MDS1, place CMP, MEGAN6, and MG-RAST results beside each other)
